# Peer review of "An Electrochemical Molecularly Imprinted Polymer Sensor for Rapid β-Lactoglobulin Detection"

_sensors, 2021, doi:10.3390/s21248240_

Round 1

Reviewer 1 Report

This work designed a molecularly imprinted polymer-modified screen-printed electrode as an electrochemical sensor for β-lactoglobulin sensing. The author also used the graphene-Au composite to enhance the conductivity. However, some important issues still need to be solved before being published.

  1. The β-lactoglobulin is a kind of protein. We know the protein has secondary structures, which are easy to change under harsh conditions, like high temperature, high pressure, low/high pH environments, etc. To keep the protein’s original structure during the imprinting process is also key research in the MIT field, like publication Eng. J. 317 (2017): 356-367. However, this work used a hydrothermal method for synthesizing MIPs. The imprinted β-lactoglobulin may not keep the original structure, thereby getting the mismatched imprinted cavities. And these obtained imprinted sites may not recognize β-lactoglobulin with high efficiency.
  2. This work used K3[Fe(CN)6] in the electrolyte. Have you considered that Fe3+ will coordinate with the protein template? Does that will affect the sensor’s results?
  3. Usually, the sensitivity of the ELISA kit cannot match that of the electrochemical sensor, which is why you design an electrochemical sensor in this work. Since their sensitivity may not be at the same level, the ELISA kit as a reference may not be used as proof for the electrochemical sensor’s results.
  4. The morphology characterizations are clear enough.
  5. For Fig. 2D, why the thin-layered composite material was replaced by the spherical structure? It should be MIP on the top of the thin-layered composite.
  6. For Fig. 2F, the author mentioned that the imprinted cavities were observed. How to prove it? Direct imaging imprinted cavities are still a hard technical difficulty in the MIT field. The cavities and rough surfaces may also cause by the elution of the template, and also can be the effect of etching, solvent/porogen, and other synthesis steps.
  7. EDS mapping is recommended to be given.
  8. There is a format error in Line 208.
  9. The peak for the DPV is not consistent with the CV. Please explain it.
  10. What is the reason for the peak shift in Fig. 6A?
  11. For the real samples experiment, the authors detected β-lactoglobulin in several brands' products. However, I believe the most important thing is to detect β-lactoglobulin in different milk species, including Whole Milk, Low fact Milk, Skim Milk.
  12. Table 2 only compared four published works. More research should be compared.
  13. This work illustrated a MIP-based screen-printed electrode. However, no special information was found in the section of the introduction. Some advanced MIP-based screen-printed electrodes should be highlighted, like Biosen. Bioelectron. 191 (2021): 113434.

Author Response

Thank you very much for the reviewers’ comments on our manuscript entitled "An Electrochemical Molecularly Imprinted Polymer Sensor for Rapid β-lactoglobulin Detection". Those comments are very helpful for revising and improving our paper. We have studied the comments carefully and made corrections which we hope to meet with approval. The main corrections are in the manuscript and the responses to the reviewers’ comments are as follows .

Point 1: The β-lactoglobulin is a kind of protein. We know the protein has secondary structures, which are easy to change under harsh conditions, like high temperature, high pressure, low/high pH environments, etc. To keep the protein’s original structure during the imprinting process is also key research in the MIT field, like publication Eng. J. 317 (2017): 356-367. However, this work used a hydrothermal method for synthesizing MIPs. The imprinted β-lactoglobulin may not keep the original structure, thereby getting the mismatched imprinted cavities. And these obtained imprinted sites may not recognize β-lactoglobulin with high efficiency.

Response 1: We heated the water bath to 90 ℃ for the synthesis of ChCl-AA deep eutectic solvent (DES), but the DES was cooled to room temperature before the addition of β-lactoglobulin. Therefore, the addition of β-lactoglobulin does not involve high temperature, high pressure and high pH. This can be confirmed in section 2.3.2 of the article.

Point 2: This work used K3[Fe(CN)6] in the electrolyte. Have you considered that Fe3+ will coordinate with the protein template? Does that will affect the sensor’s results?

Response 2: Usually potassium ferricyanide in electrochemical sensors is used as a redox probe. That means it would not react with β-lactoglobulin. Potassium ferricyanide was also used as a REDOX probe in the study of lactoglobulin detection by Eissa et al.

References:

[1] Eissa, S., Tlili, C., L'Hocine, L., & Zourob, M. (2012). Electrochemical immunosensor for the milk allergen β-lactoglobulin based on electrografting of organic film on graphene modified screen-printed carbon electrodes. Biosensors and Bioelectronics, 38(1), 308-313.

Point 3: Usually, the sensitivity of the ELISA kit cannot match that of the electrochemical sensor, which is why you design an electrochemical sensor in this work. Since their sensitivity may not be at the same level, the ELISA kit as a reference may not be used as proof for the electrochemical sensor’s results.

Response 3: ELISA is the most common method to detect β-lactoglobulin for its advantages in accuracy. ELISA kits were also used for comparison in the studies of Montiel et al and Eissa et al.

References:

[1] Montiel, V. R. V., Campuzano, S., Conzuelo, F., Torrente-Rodríguez, R. M., Gamella, M., Reviejo, A. J., & Pingarrón, J. M. (2015). Electrochemical magnetoimmunosensing platform for determination of the milk allergen β-lactoglobulin. Talanta, 131, 156-162.

[2] Eissa, S., Tlili, C., L'Hocine, L., & Zourob, M. (2012). Electrochemical immunosensor for the milk allergen β-lactoglobulin based on electrografting of organic film on graphene modified screen-printed carbon electrodes. Biosensors and Bioelectronics, 38(1), 308-313.

Point 4: The morphology characterizations are clear enough.

Response 4: We have replaced Figure 2 with a higher resolution image and will pay more attention to image sharpness in future posts.

Point 5: For Fig. 2D, why the thin-layered composite material was replaced by the spherical structure? It should be MIP on the top of the thin-layered composite.

Response 5: According to the reviewer’s suggestion, we have corrected the “PEI-rGO-Au-NCs@MIP” into “MIP” in line 178 of the revised manuscript.

Point 6: For Fig. 2F, the author mentioned that the imprinted cavities were observed. How to prove it? Direct imaging imprinted cavities are still a hard technical difficulty in the MIT field. The cavities and rough surfaces may also cause by the elution of the template, and also can be the effect of etching, solvent/porogen, and other synthesis steps.

Response 6: According to the reviewer’s suggestion, We have revised our conclusion in line 171 to 173 of the revised manuscript.

Point 7: EDS mapping is recommended to be given.

Response 7: EDS mapping is a visual way to tell both the morphology and elements distribution of the synthesized material. However, we have got the structure and element composition by SEM, EDS and infrared spectroscopy.

Point 8: There is a format error in Line 208.

Response 8: We have changed it to the correct format in the revised manuscript.

Point 9: The peak for the DPV is not consistent with the CV. Please explain it.

Response 9: CV is generally used to show that the target substance can produce an oxidation-reduction reaction on the electrode for qualitative. The responsiveness of DPV is greater than CV, so it is usually used to determine the content of the target substance. The responsiveness of DPV is higher than CV, so the peak value is different.

Point 10: What is the reason for the peak shift in Fig. 6A?

Response 10: The DPV peak shift phenomenon usually occurs in the study of screen printing electrode, but it does not affect the detection result of the target substance.

Point 11: For the real samples experiment, the authors detected β-lactoglobulin in several brands' products. However, I believe the most important thing is to detect β-lactoglobulin in different milk species, including Whole Milk, Low fact Milk, Skim Milk.

Response 11: Whole milk, skimmed milk and low-fat milk differentiate each other in fat content, rather than protein composition. Therefore, it is not necessary to detect β-lactoglobulin in these three types of milk.

Point 12: Table 2 only compared four published works. More research should be compared.

Response 12: According to the reviewer’s suggestion, we added the detection range and detection limit of ELISA and SPR in Table 2.

Point 13: This work illustrated a MIP-based screen-printed electrode. However, no special information was found in the section of the introduction. Some advanced MIP-based screen-printed electrodes should be highlighted, like Biosen. Bioelectron. 191 (2021): 113434.

Response 13: We have highlighted excellent MIP screen printed electrodes in the introduction section. This can be found on line 55-58 of the revised manuscript.

Reviewer 2 Report

This manuscript reports a novel material for the electrochemical sensor targeting the β-Lg, which is important for the milk industry and people who have allergies. The sensor was well designed and show very competitive performance in dynamic range, selectivity and limit of detection. Overall, the manuscript have good scientific soundness, the experiments were well designed with proper control. However, there are some minor issues that need to be addressed.

  1. the manuscript has some issue in grammar and the choice of words. Please edit it with care or consider a professional editing service.
  2. reference 9 seems to be about detecting the source of the milk, rather than detecting the β-Lg. PCR is a widely used method for detecting DNA, it probably won’t be a good method to test proteins. Please provide more literature about this if the author still wants to make the claim that PCR is a method for detection of β-Lg.
  3. In the introduction part, please include a concentration range for the usual milk sample.
  4. In Table 1, it seems most of the real-world samples would have β-Lg concentration >10 ug/mL, or 10E-2 mg/mL. But in the standard curve, the author only measured up to 10E-4 mg/mL. Is there a reason why it doesn’t go any higher?  Reading beyond the range of standard curve would not give accurate data. Sometimes it might give very large error if it’s no longer linear in that range.
  5. In table 2, if the author can also give the detection range and limit of Elisa kit and other reported methods, that would be clearer on how the sensor performs among all the available methods.
  6. In figure 6A, the text on the figure is 0 -> 10^-9 mg/mL. Apparently, that’s not correct. There is also no legend to show which color is which concentration.
  7. In figure 6B, the font size of the label (Lg C β-Lg ) is too small, the ‘-9’ is very difficult to read.
  8. In figure 7 and section 3.6, the concentrations of the proteins were not described.
  9. the Abbreviations table is not exclusive, for example it doesn’t have the SPCE used in this work.

Author Response

Thank you very much for the reviewers’ comments on our manuscript entitled "An Electrochemical Molecularly Imprinted Polymer Sensor for Rapid β-lactoglobulin Detection". Those comments are very helpful for revising and improving our paper. We have studied the comments carefully and made corrections which we hope to meet with approval. The main corrections are in the manuscript and the responses to the reviewers’ comments are as follows.

Point 1: The manuscript has some issue in grammar and the choice of words. Please edit it with care or consider a professional editing service.

Response 1: According to the reviewer’s advice, we have embellished the article by changing some words and sentences.

Point 2: Reference 9 seems to be about detecting the source of the milk, rather than detecting the β-Lg. PCR is a widely used method for detecting DNA, it probably won’t be a good method to test proteins. Please provide more literature about this if the author still wants to make the claim that PCR is a method for detection of β-Lg.

Response 2: Due to our misunderstanding of the literature, we have deleted this part of the article on line 42.

Point 3: In the introduction part, please include a concentration range for the usual milk sample.

Response 3: We have increased the amount of β-lactoglobulin in regular milk in line 37-38 of the revised manuscript.

Point 4: In Table 1, it seems most of the real-world samples would have β-Lg concentration >10 ug/mL, or 10E-2 mg/mL. But in the standard curve, the author only measured up to 10E-4 mg/mL. Is there a reason why it doesn’t go any higher ? Reading beyond the range of standard curve would not give accurate data. Sometimes it might give very large error if it’s no longer linear in that range.

Response 4: We diluted the milk samples to an appropriate concentration before employing electrochemical sensors for tests.

Point 5: In table 2, if the author can also give the detection range and limit of Elisa kit and other reported methods, that would be clearer on how the sensor performs among all the available method

Response 5: According to the reviewer’s suggestion, we added the detection range and detection limit of ELISA and SPR in Table 2.

Point 6: In figure 6A, the text on the figure is 0 > 10^-9 mg/mL. Apparently, that’s not correct. There is also no legend to show which color is which concentration.

Response 6: We have added the legend to show the color of β-Lg concentration in the title of Figure 6A in Line 248 in the revised version manuscript, and modified the correct concentration range to 0-10-4.

Point 7: In figure 6B, the font size of the label (Lg C β-Lg ) is too small, the ‘-9’ is very difficult to read.

Response 7: Following the reviewer’s suggestion, we have increased the font size of Figure 6B in line 248 of the revised manuscript.

Point 8: In figure 7 and section 3.6, the concentrations of the proteins were not described.

Response 8: Following the reviewer’s suggestion, we have added the concentration of interfering substances in lines 255-257 of the revised manuscript.

Point 9: The Abbreviations table is not exclusive, for example it doesn’t have the SPCE used in this work.

Response 9: We have added the full name of SPCE to the table.

Round 2

Reviewer 1 Report

The author's answers are well. I recommend accepting this manuscript in this version.

Author Response

Thanks very much for the suggestions of the reviewer. We have corrected some spelling and grammar errors in the article.